# Regulatory Ability of *Lactiplantibacillus plantarum* on Human Skin Health by Counteracting In Vitro *Malassezia furfur* Effects

**DOI:** 10.3390/jof9121153

**Published:** 2023-11-29

**Authors:** Alessandra Fusco, Brunella Perfetto, Vittoria Savio, Adriana Chiaromonte, Giovanna Torelli, Giovanna Donnarumma, Adone Baroni

**Affiliations:** 1Department of Experimental Medicine, Section of Microbiology and Clinical Microbiology, University of Campania “Luigi Vanvitelli”, 80138 Naples, Italy; brunella.perfetto@unicampania.it (B.P.); vittoriasavio@libero.it (V.S.); adrianachiaromonte1993@gmail.com (A.C.); giovannatorelli@libero.it (G.T.); 2Department of Mental and Physical Health and Preventive Medicine, University of Campania “Luigi Vanvitelli”, 80138 Naples, Italy; adone.baroni@unicampania.it

**Keywords:** *Malassezia furfur*, *Lactiplantibacillus plantarum*, skin barrier, inflammasome, aryl hydrocarbon receptor, Nrf-2

## Abstract

The skin serves as the first barrier against pathogen attacks, thanks to its multifunctional microbial community. *Malassezia furfur* is a commensal organism of normal cutaneous microflora but is also a cause of skin diseases. It acts on different cell pattern recognition receptors (TLRs, AhR, NLRP3 inflammasome) leading to cellular damage, barrier impairment, and inflammatory cytokines production. *Lactobacillus* spp. Is an endogenous inhabitant of healthy skin, and studies have proven its beneficial role in wound healing, skin inflammation, and protection against pathogen infections. The aim of our study is to demonstrate the ability of live *Lactiplantibacillus plantarum* to interfere with the harmful effects of the yeast on human keratinocytes (HaCat) in vitro. To enable this, the cells were treated with *M. furfur*, either alone or in the presence of *L. plantarum*. To study the inflammasome activation, cells require a stimulus triggering inflammation (LPS) before *M. furfur* infection, with or without *L. plantarum. L. plantarum* effectively counteracts all the harmful strategies of yeast, reducing the phospholipase activity, accelerating wound repair, restoring barrier integrity, reducing AhR and NLRP3 inflammasome activation, and, consequently, releasing inflammatory cytokines. Although lactobacilli have a long history of use in fermented foods, it can be speculated that they can also have health-promoting activities when topically applied.

## 1. Introduction

To maintain healthy skin function, it is important to have a balanced skin microbiota, as alterations in its integrity are linked to several cutaneous diseases [1,2]. The microbes present on the skin form a barrier capable of counteracting environmental colonization through various mechanisms, including resource exclusion, direct inhibition, and/or interference with the colonization. The cutaneous microbiota improves the physical skin barrier, contributing to the differentiation and epithelialization, stimulating the innate and adaptive immunity, releasing antimicrobial peptides, and influencing the development of protective immunity [1,2]. Recent research has emphasized the role of functional probiotics in intestinal function and the maintenance of healthy skin [3,4,5,6]. However, the mechanism of the interactions between the maintenance of the skin barrier and probiotics has not been elucidated yet.

The yeasts of the genus *Malassezia* colonize human skin after birth. They are saprophytes and, as such, are generally well-tolerated by the human immune system. However, under appropriate conditions, they can invade the stratum corneum and interact with the host immune system, developing their pathogenic potential. Within the genus *Malassezia*, *Malassezia furfur* is involved in skin diseases of varying severity, such as pityriasis versicolor, seborrheic dermatitis, atopic eczema, dermatitis, and dandruff [7,8,9]. It is very probable that the inflammatory effects exerted on the host cells by *Malassezia* and the impairment of the epithelial barrier result from a complicated balance involving different molecules. The expression of these molecules is mediated not by one, but by different pathways involving different receptors, cofactors, and transcription factors.

It has been shown that the triggering of some skin inflammatory processes is linked to the response of mast cells and other cellular types to stimulation by *Malassezia* spp. through the canonical Toll-like receptor 2 (TLR2)/MyD88 pathway. This stimulation leads to either the up- or downregulation of cytokine, chemokine, and adhesion molecules in host effector cells. Following the triggering of inflammation, metalloproteinases and tight junctions can be modulated, too [10,11].

For the control of membrane permeability and integrity, crucial structures are the tight junctions, including occludin, claudins (integral membrane proteins), and the junction adhesion molecules that interact with the zonula occludens (ZO-1, ZO-2, and ZO-3). Additionally, growth factors, extracellular matrix components (ECM), and the metalloproteinase family (MMPs) play an important role in the repair of damaged skin. Among the MMPs, type IV collagenases such as MMP2 and 9, released by keratinocytes, have catalytic effects on laminin, collagens, and gelatin, thereby playing an important role in remodeling the normal architecture of the damaged tissue [12,13,14].

The virulence factors of various pathogens, including secretory hydrolytic enzymes, proteinases, lipolytic enzymes, lipases, and phospholipases, can destabilize the host cell membrane. The phospholipase of *Malassezia* spp. is one of the different factors responsible for skin barrier impairment during the complex interaction between yeasts and the host [15,16].

*Malassezia* yeasts can also synthesize several indolic compounds that act as potent ligands for a nuclear receptor the aryl hydrocarbon (AhR), which has pluripotent biological functions in the skin and other organs. After activation, AhR in the nucleus dimerizes with the aryl hydrocarbon nuclear translocator (ARNT). This complex Ahr/ARNT is able to link to xenobiotic responsive elements (XRES), increasing the transcription of several genes and the expression of the associated proteins such as CYP1A1, OVOL1, filaggrin (FLG), loricrin (LOR), and involucrin (INV) [17,18,19]. Mutagen metabolites and reactive oxygen species (ROS), by activating AhR, lead to the expression of CYP1A1, a functional biomarker present in mammal cells, responsible for the activation of the AhR-mediated signaling pathway [20].

The innate immune response serves as the first line of defense against different pathogens. Pathogen-associated molecular patterns (PAMPS) bind with the pattern recognition receptors (PRR) on the cell surface, with some PRRs capable of binding specific fungal cell wall components of *Malassezia*. Among these PRRs are the inflammasomes, which include: (i) sensor proteins, such as Nod-like receptors (NLRP1, NLRP3, NLRC4, and AIM2); (ii) an adaptor protein, such as an apoptosis-associated speck-like protein containing C-terminal caspase recruitment domain (CARD) (ASC); (iii) an effector protease protein, such as caspase-1. The assembly of inflammasomes and the activation of caspase-1 are responsible for cleaving and activating inflammatory cytokines pre-forms, including pro-IL-1β and pro-IL-18, into mature forms. Their subsequent release induces an inflammatory response [21,22,23].

Over the past few decades, live microorganisms defined as probiotics, including *Bifidobacterium*, *Saccharomyces*, *Enterococcus*, *Bacillus*, and *Lactobacillus*, have been successfully used to improve human health by regulating the immune response. Additionally, different products of bacterial derivation, particularly those released during the fermentation processes of food matrices or in culture, known as post-biotics, have been studied for their anti-inflammatory properties. Recently, Lactic bacilli have been extensively studied, evaluating their beneficial effects on the skin. It has been demonstrated how *Lactiplantibacillus plantarun* can act to alleviate atopic dermatitis symptoms in adults, due to its immunomodulatory effects, or can act against UVB damage in dermal fibroblasts and hairless mice [24,25,26,27]. Karczewski J. et al. showed the protective effect of *L. plantarum* on epithelial barriers in vivo [12]. Jeong JH et al. demonstrated the beneficial effects of *L. plantarum* on the intestine and also the effects of *L*. *plantarum* LTA as anti-photoaging on the skin by regulating the expression of MMP1 through the intake of “beneficial food” [28]. Brandi et al. showed that Lactobacilli lysates are able to induce the dysregulation of proteins such as interleukins in keratinocytes and activate some transcription factors such as NFKB [5]. In addition, soluble fractions of some lactobacilli strains have also been correlated with the modulation of nitric oxide synthase 2 (NOS2) in HaCat cells by Lombardi et al. [6].

The aim of our study is to demonstrate that the presence of *L. plantarum* is able to interfere with the harmful effects of *M. furfur* in an in vitro experimental model using human keratinocyte HaCat cells.

## 2. Materials and Methods

### 2.1. Cell Cultures

The immortalized human keratinocyte HaCat cell line (Elabsciences, Houston, TX, USA) was cultured in Dulbecco’s Modified Eagle Medium (DMEM–Gibco, Waltham, MA, USA), supplemented with 1% Penstrep, 1% glutamine, and 10% fetal calf serum (Invitrogen, Carlsbad, CA, USA) at 37 °C in air and 5% CO_2_. Before performing the experiments, the cells were seeded in 6-well plates until reaching 80% confluence.

### 2.2. Microorganisms and Culture Media

*M. furfur* ATCC^®^ 14521, obtained from the American Type Culture Collection (Rockville, MD, USA), was grown for four days at 30 °C in Sabouraud dextrose agar (SDA-Oxoid, Milan, Italy) containing peptone (1%), glucose (4%), olive oil (2%), and Tween 80 (0.2%). The cells were collected in PBS, separated from the medium by centrifugation at 2800× *g* for 5 min, washed twice with PBS, gently vortexed to avoid yeast aggregation, and re-suspended in DMEM. For each experiment, a ratio of 30:1 yeasts/cell was used. *L. plantarum* (ATCC^®^ 8014) was cultured in Man, Rogosa, and Sharpe medium (MRS-Oxoid, Milan, Italy) at 37 °C under microaerophilic conditions.

### 2.3. Phospholipase Assay

To evaluate the probable effects of *L. plantarum* on *M. furfur* phospholipase activity, we carried out a co-culture of *L. plantarum* and *M. furfur,* both at 0.3 O.D. (approximatively 10^6^ CFUs/mL) in MRS broth at 30 °C for 48 h. As a positive test control, we also inoculated *M. furfur* alone in MRS for 48 h to ensure the viability of *M. furfur* in this culture medium. At the end of this time, *M. furfur* was placed on SDA containing peptone (1%), glucose (4%), olive oil (2%), and Tween 80 (0.2%), with the addition of 10 mM of β-endorphin (Merck, Darmstadt, Germany), and was incubated at 30 °C for 10 days. Subsequently, four individual colonies of the yeast were picked, transferred to egg-yolk agar (SDA containing 1 M sodium chloride, 0.005 M calcium chloride, and 10% sterile egg yolk), and incubated for an additional 10 days at 30 °C [29,30,31].

### 2.4. Cell Treatments

For the evaluation of barrier integrity and AhR pathway, semi-confluent cell monolayers were treated with *M. furfur* at a ratio of 30:1 yeasts/cell for 24 and 48 h. Alternatively, before *M. furfur* addition, HaCat cells were pre-treated for 2 h with *L. plantarum* (~10^8^ CFUs/mL) at a multiplicity of infection (MOI) of 100, at 37 °C at 5% CO_2_ in DMEM without antibiotics. For inflammasome activation, cells were pre-treated with LPS of *Salmonella enterica* subsp. *enterica* serovar Typhimurium (Merck, Darmstadt, Germany) at a concentration of 100 ng/mL for 4 h [23]. Subsequently, they were infected with *M. furfur* (30:1 yeasts/cell) alone or in the presence of *L. plantarum* (~10^8^ CFUs/mL) for 4 h at 37 °C at 5% CO_2_.

### 2.5. Real-Time PCR 

At the end of the experiments, the total mRNA was extracted from the HaCat cells. Further, 500 ng of the mRNA was reverse-transcribed (Expand Reverse Transcriptase- Roche, Monza, Italy) into complementary DNA (cDNA) using random hexamer primers (Random hexamers-Roche) at 42 °C for 45 min, according to the manufacturer’s instructions. Real-time PCR for *AhR*, *CyP1A1*, *Occludin*, *Zonulin-1*, *Claudin-1*, *Filaggrin*, *Nrf-2*, *MMP-2*, *Caspase-1*, *IL-1 β*, *IL-18,* and *NLRP-3* was carried out with the LC Fast Start DNA Master SYBR Green kit using 2 μL of cDNA corresponding to 40 ng of the total RNA for a final volume of 20 μL, 3 μM MgCl_2_, and 0.5 μM sense and antisense primers (Table 1). At the end of each run, the melting curve profiles were obtained by cooling the sample to 65 °C for 15 s and then slowly heating it at 0.20 °C/s up to 95 °C with continuous measurements of the fluorescence to confirm the amplification of specific transcripts. Cycle-to-cycle fluorescence emission readings were monitored and analyzed using LightCycler^®^ 2.0 Software (Roche Diagnostics, Monza, Italy). Melting curves were generated after each run to confirm the amplification of specific transcripts. The *β-actin* coding gene, one of the most commonly used housekeeping genes, was used as the internal control gene. All reactions were carried out in triplicate, and the relative expression of a specific mRNA was determined by calculating the fold change relative to the *β-actin* control. The fold change of the tested gene mRNA was obtained using the Roche 2.0 Software for the amplification efficiency of each primer, as calculated by the dilution curve. The specificities of the amplification products were verified by subjecting the amplification products to electrophoresis on 1.5% agarose gel and visualization with ethidium bromide staining [32].

### 2.6. ELISA Assay

The presence of AhR, MMP-2, Caspase-1, IL-1β, IL-18 (Elabsciences Biotechnology Inc., Houston, TX, USA), and Nrf-2 (ThermoFisher Scientific, Waltham, MA, USA) was detected in cell supernatants, while Zonulin-1, Occludin, Claudin-1, (Elabscience Biotechnology Inc.), and Filaggrin (Cusabio, Houston, TX, USA) were detected in lysates of HaCat cells infected with *M. furfur* and *L. plantarum*, with or without LPS, using enzyme-linked immuno-sorbent assay (ELISA), according to the manufacturer’s instructions.

### 2.7. Scratch Wound Healing Assay

For scratch wound healing assay, HaCat cells were seeded in 12-well tissue culture plates in DMEM medium supplemented with 1% glutamine and 10% fetal calf serum without antibiotics at 37 °C in 5% CO_2_. When the cells reached a confluence of ~80%, *L. plantarum* (~10^8^ CFUs/mL), at a multiplicity of infection (MOI) of 100:1 cell, was added. After 6 h of incubation, a scratch was made on the monolayer using a sterile p200 pipette tip. Cells were then slightly washed with PBS to remove debris, the medium was replaced with fresh DMEM medium, and the cells were infected with *M. furfur* at a ratio of 30:1 yeasts/cell. The scratch was monitored until 48 h were completed, comparing the area of the scratch of the control cells with those treated with *Malassezia* alone or with the pretreatment of *L. plantarum*.

### 2.8. Statistical Analysis

Significant differences among groups were assessed through two-way ANOVA by using GraphPad Prism 8.0, and the comparison between the means was calculated using student’s *t*-test. The data are expressed as means ± standard deviation (SD) of three independent experiments.

## 3. Results

### 3.1. Effect of L. plantarum on M. furfur Phospholipase Activity

To assess the influence of the *L. plantarum* on the expression of *M. furfur*’s virulence factors, the phospholipase activity assay was carried out by incubating *M. furfur* for 48 h with *L. plantarum*. Subsequently, *M. furfur* was incubated first on the medium with β-endorphin for 10 days and then on egg-yolk agar for an additional 10 days. The phospholipase activity was revealed by the appearance of a precipitation halo around the colonies, resulting from the release of calcium and fatty acids due to the degradation of the phospholipids in the egg yolk. As shown in Figure 1, the phospholipase activity is somewhat reduced after incubation with *L. plantarum*.

### 3.2. Effect of L. plantarum on Barrier Integrity

The ability of *L. plantarum* to strengthen the epithelial barrier was evaluated by analyzing the expression of the genes encoding the TJs: Occludin, Zonulin, and Claudin-1, by using real-time PCR (Figure 2a–c) and measuring the production of the corresponding proteins by ELISA assay (Figure 2d–f). The data obtained indicate that *M. furfur* alone damages the epithelial barrier, leading to the consequent reduction of TJs expression. However, pretreatment with *L. plantarum* before infection improves the barrier conditions, maintaining levels of Claudin-1 and Zonulin (up to 24 h of infection), similar to the uninfected control. Meanwhile, it has no effect on the expression of Occludin.

### 3.3. Tissue Repair Activity of L. plantarum

The ability of *L. plantarum* to promote tissue repair was evaluated by wound healing assay and by analyzing the induction of MMP-2 expression. As shown in Figure 3, the presence of *L. plantarum* promotes wound healing, which is inhibited in the presence of *M. furfur* alone. Furthermore (Figure 4), the presence of *L. plantarum* stimulates the production of MMP-2, reduced by *M. furfur*.

### 3.4. Activation of AhR Pathways

The analysis of AhR pathways activation demonstrated that *M. furfur* is able to activate this receptor (Figure 5a,b) and Cyp1A1 (Figure 5c) after 48 h of treatment. In addition; the downregulation of Nrf-2 expression caused by *M. furfur* infection is counteracted by the activity of *L. plantarum* after 48 h (Figure 6a,b), while the Filaggrin-1 expression was enhanced (Figure 6c,d), both at 24 and 48 h. Following pretreatment, *L. plantarum* inhibits the expression of AhR and CyP1A1 induced by *M.furfur*, keeping high Filaggrin expression. 

### 3.5. Inflammasome Activation

The analysis of inflammasome activation was carried out by cell pretreatment with *S.* Typhimurium LPS and then adding *M. furfur*, either alone or in presence of *L. plantarum*. The data obtained (Figure 7 and Figure 8) show that *M. furfur* activates the inflammasome by inducing the expression of Il-1β, Caspase-1, IL-18, and NLRP-3 after 4 h of infection, while the simultaneous addition of *L. plantarum* totally inhibits this activation, bringing them back to the control values.

## 4. Discussion

Commensal bacteria are beneficial for skin health, acting on the host immune system and counteracting the attack of pathogenic strains. Lactobacilli have often been used in fermented foods due to their ability to colonize various body districts, such as the gastrointestinal tract, urogenital tract, and nasal cavity. It is not well-known whether lactobacilli could also colonize and promote skin health. Sarah Lebeer et al. [33] demonstrated that taxa of *Lactobacillus,* typically associated with the human vagina, were also present on the skin as the most prevalent species. Additionally, nomadic or free-living lactobacilli, including *L. plantarum,* are frequently detected, suggesting that some lactobacilli on the skin could also result from fermented food sources and thus be transient passengers.

*Malassezia* spp. also exist as commensals in normal cutaneous microflora, but can cause dandruff, seborrheic dermatitis, pityriasis versicolor, and onycomycosis, and can exacerbate atopic dermatitis and psoriasis. At the basis of the double-faced commensal vs pathogens, there are probably changes in both the fungus (e.g., variable secretion of AhR agonist) and the host conditions [34]. In addition, *M. furfur* might be an excellent model for studying how the activation of multiple pattern recognition receptors (TLRs, AhR pathway, NLRP3 inflammasome) and the induction of several cells’ regulatory molecules coordinate the different responses of cells in contact with yeast [19,35,36,37,38]. Here, we demonstrate how *L. plantarum* can counteract the effects of *M. furfur* induced in human keratinocytes (HaCat) through different pathways of activation. Initially, *L. plantarum* can be beneficial for skin colonized by *M. furfur* by reducing the production of phospholipase, considered a *Malassezia* virulence factor, and promoting healing of wounds inhibited in the presence of lipophilic yeast. Additionally, it acts to limit barrier damage. Chronic skin inflammation is often associated with changes in epithelial permeability, as confirmed by studies on the expression of several claudins, measurements of transepithelial electrical resistance, and experiments with RNA interference in polarized gut epithelial cell lines. These permeability disfunctions are consistently associated with alterations of TJs [12]. We demonstrated that *M. furfur* reduces the levels of occludin, zonulin, and claudin in HaCat cells in vitro. *L. plantarum* pretreatment is able to restore levels of zonulin and claudin back to, and beyond, those of the control at 24 and 48 h, respectively. However, it does not act on the reduction of occludin levels induced by *M. furfur*. *L. plantarum* also increases the level of MMP2, both at 24 and 48 h, which was downregulated by *Malassezia*, probably related to the increased level of claudin. Recently, Oku et al. demonstrated that, during the granulation phase, there are high levels of MMP-2, which remain elevated throughout the entire remodeling phase of ECM. This evidence may be responsible for keratinocytes’ migration and has been linked to an increased expression of Claudin-1 [14]. Keratinocyte differentiation is regulated by several factors of transcription, such as AHR, OVOL1/2, MYC, NOTCH1, CEBP, and PPAR. Inflammatory cytokines, phytochemicals, and UV-mediated oxidative stress are able to activate or modulate these transcription factors. For instance, IL-4 and IL-13, pathogenic for atopic dermatitis, and IL-17A, pathogenic for psoriasis, lead to a downregulation of filaggrin. This contributes to the dry, barrier-impaired skin lesions in these pathologies. In some cases, simultaneously with the activation of AhR, antioxidant ligands activate the phosphorylation of Nrf2, which translocates to the nucleus, binds small MAf proteins, recognizes antioxidant-responsive elements, and induces the expression of target genes [17,18,25,39]. Here, we report that *M. furfur* treatment induces an activation of AhR, CyP1A1, and a consequent increase in filaggrin expression at 48 h. The presence of *L. plantarum* reduces AhR and CyP1A1 activation at 48 h, enhances the NrF-2 levels downregulated by *Malassezia*, and maintains high levels of filaggrin.

Previous reports have shown that different fungi, including some strains of *Malassezia* spp., are able to activate the NLRP3 inflammasome, leading to robust secretion of the pro-inflammatory cytokine IL-1β and IL18 in human antigen-presenting cells or in other cell types, favoring cell colonization and chronic skin inflammation. In contrast, *M. furfur* seems unable to activate the pathway NLRP3 inflammasome in human keratinocytes but is necessary a triggering stimulus such as a PAMPS, the LPS [21,23,40]. In this paper, we demonstrated that *M. furfur* activation of the NRLP inflammasome in HaCat cells caused an increase of NLRP3, Caspase-1, and subsequently increased levels of IL-1β and IL-18. However, the presence of *L. plantarum* reduces the expression of the inflammatory signals triggered by LPS and carried forward by *M. furfur* infection.

In fungal infections, the most common therapeutic options include topical and systemic agents. The poor effects of topical antimycotics, the need for long treatments with consequent damage to the liver and kidneys, and the emergence of resistant variants limit the therapeutic efficacy of current antifungal drugs. The need to resort to alternative therapies has stimulated both basic research and clinical studies, such as the use of lactobacilli in topical formulations [33]. Studies have been conducted on acne, atopic dermatitis, and rosacea [41,42].

With the limitation of in vitro experimental models, our results strongly support the hypothesis that *L. plantarum,* as a probiotic, can improve skin health. Particularly in our study, *L. plantarum,* used as live bacteria in cell pretreatment before *Malassezia* infection, reduces yeast growth in culture, reduces its phospholipase activity, and promotes the repair of tissue damage slowed down by *Malassezia*. In addition, it blocks the membrane alterations in human keratinocytes infected with *M. furfur*, modulating the TJs and the AhR-Nrf2-mediated responses, and reduces the inflammatory process mediated by NLR3-inflammasome, IL-1β, induced in HaCat cells by *M. furfur* infection. Our experimental model of pretreatment tries to mimic a condition that predicts, in vivo, the effect of *Malassezia* spp. on the skin of a host who uses a pharmacobiotic as a protective agent (such as cosmetics, drugs, or “beneficial food”), speculating that this pharmacobiotic can also be used post-infection in combination with traditional antifungal therapies, thanks to its ability to manipulate the cutaneous microbiome and promote skin health. This expands the spectrum of available treatment options for topical applications in both aesthetic and regenerative medicine.

## Figures and Tables

**Figure 1 jof-09-01153-f001:**
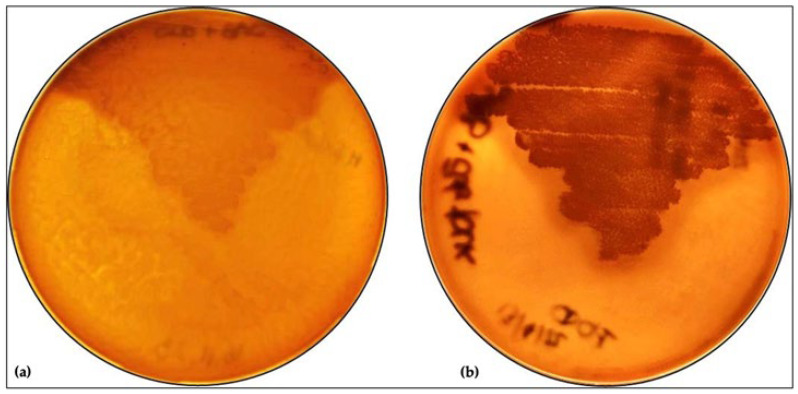
Egg yolk agar with *M. furfur* pretreated for 48 h, either with *L. plantarum* (**a**) or alone (**b**). (scale 1:0.8).

**Figure 2 jof-09-01153-f002:**
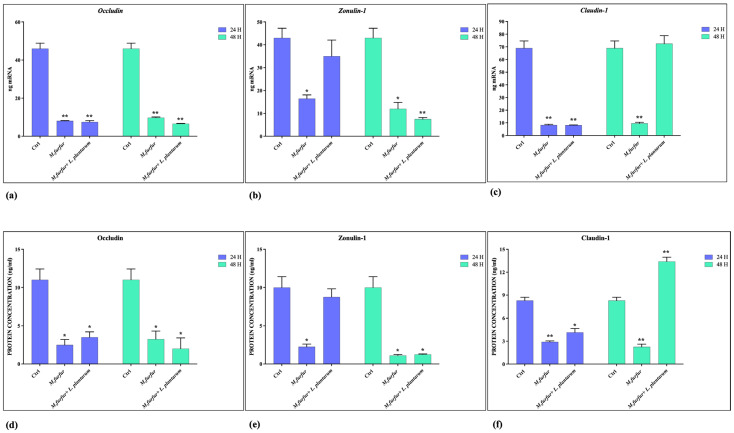
Tight junction expression analysis. (**a**–**c**) mRNA expression of Occludin, Zonulin, and Claudin-1 in HaCat cells treated with *M. furfur* (30:1 MOI), either alone or pretreated for 2 h with *L. plantarum* (100:1 MOI) after 24 and 48 h; (**d**–**f**) Protein expression of Occludin, Zonulin, and Claudin-1 in the same experimental conditions and for the same time periods. The data are representative of three different experiments ± SD. Significant differences are indicated by * *p* < 0.05, ** *p* < 0.01.

**Figure 3 jof-09-01153-f003:**
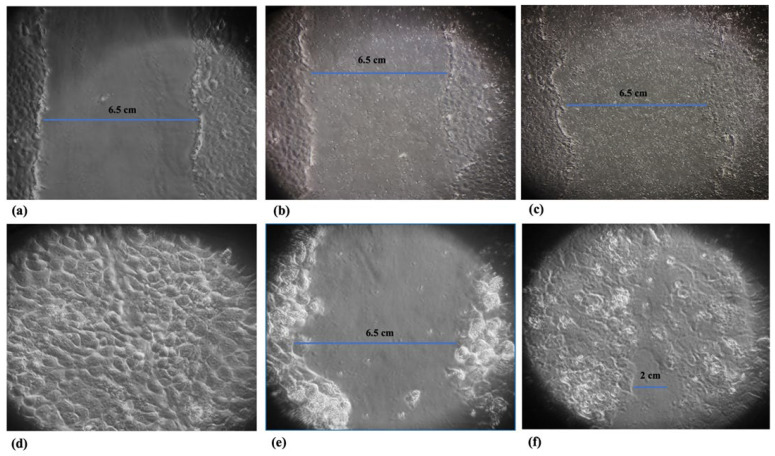
Scratch wound healing assay: (**a**–**c**) Time 0, magnification 10×; Control cell, HaCat treated with *M. furfur* (30:1 yeasts/cell), and HaCat infected with *M.furfur* (30:1 yeasts/cell) with pretreatment for 6 h with *L. plantarum* (100:1 bacteria/cell), respectively; (**d**–**f**) The same samples after 48 h of incubation, magnification 20×.

**Figure 4 jof-09-01153-f004:**
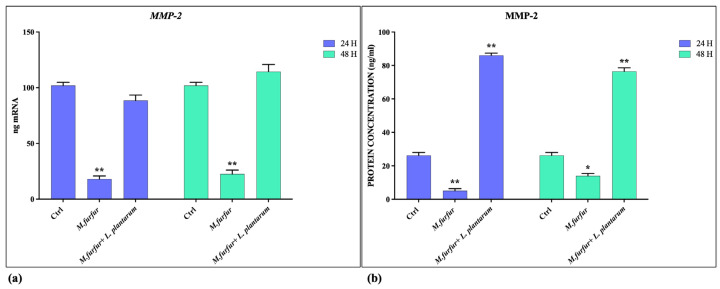
Metalloproteinase expression evaluation. mRNA (**a**) and protein (**b**) expression of MMP2 in HaCat cell treated with *M. furfur* (30:1 MOI), either alone or pretreated for 2 h with *L. plantarum* (100:1 MOI) after 24 and 48 h. The data are representative of three different experiments ± SD. Significant differences are indicated by * *p* < 0.05, ** *p* < 0.01.

**Figure 5 jof-09-01153-f005:**
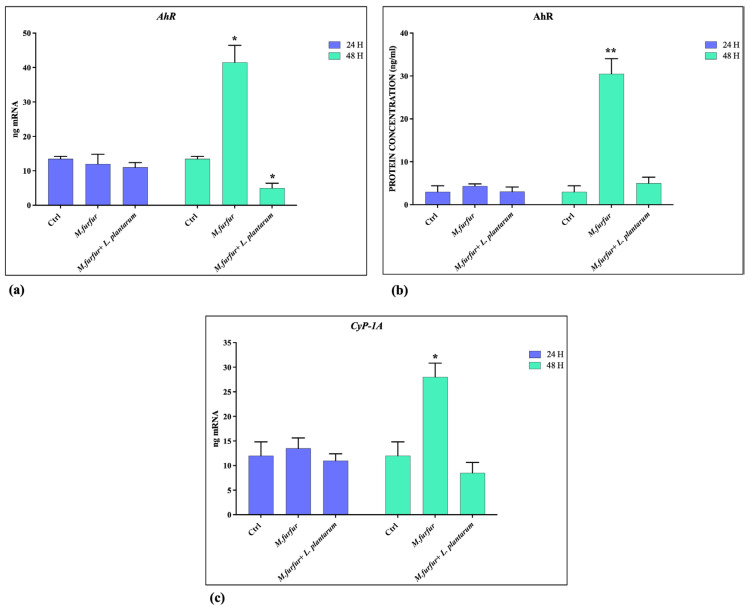
AhR pathway activation analysis. mRNA (**a**,**b**) and protein (**c**) expression of AhR and Cyp-1A in HaCat cell treated with *M. furfur* (30:1 MOI), either alone or pretreated for 2 h with *L. plantarum* (100:1 MOI) after 24 and 48 h. The data are representative of three different experiments ± SD. Significant differences are indicated by * *p* < 0.05, ** *p* < 0.01.

**Figure 6 jof-09-01153-f006:**
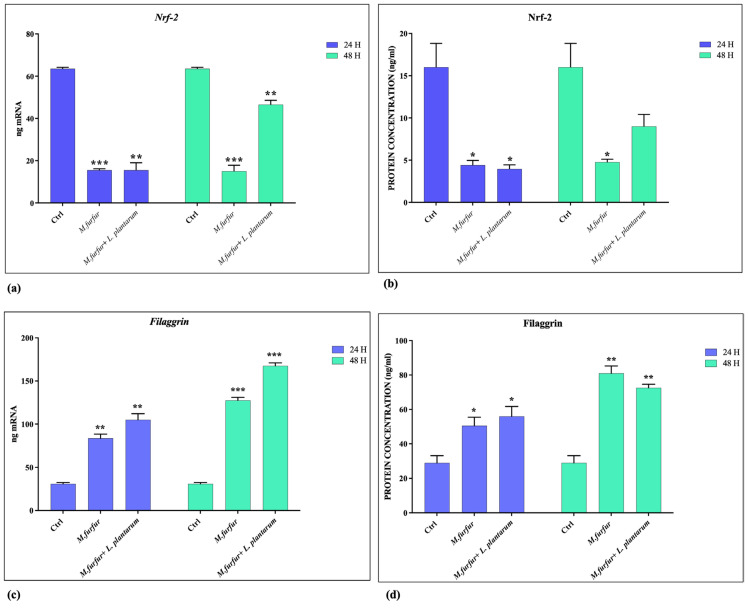
Nrf-2 and Filaggrin activation analysis. mRNA (**a**,**c**) and protein (**b**,**d**) expression of Nrf-2 and Filaggrin in HaCat cell treated with *M. furfur* (30:1 MOI), either alone or pretreated for 2 h with *L. plantarum* (100:1 MOI) after 24 and 48 h; The data are representative of three different experiments ± SD. Significant differences are indicated by * *p* < 0.05, ** *p* < 0.01, *** *p* < 0.001.

**Figure 7 jof-09-01153-f007:**
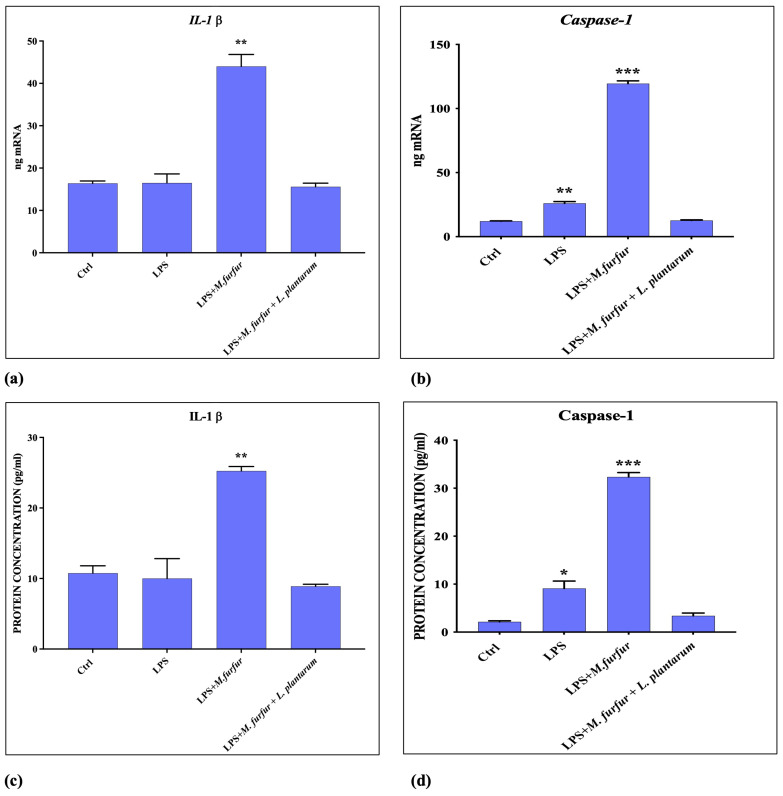
Inflammasome activation evaluation. mRNA (**a**,**b**) and protein (**c**,**d**) expression of IL-1β and Caspase-1 in HaCat cell pre-treated for 4 h with LPS of *S.* Typhimurium and infected with *M. furfur* alone or in presence of *L. plantarum* after 24 and 48 h. The data are representative of three different experiments ± SD. Significant differences are indicated by * *p* < 0.05, ** *p* < 0.01, *** *p* < 0.001.

**Figure 8 jof-09-01153-f008:**
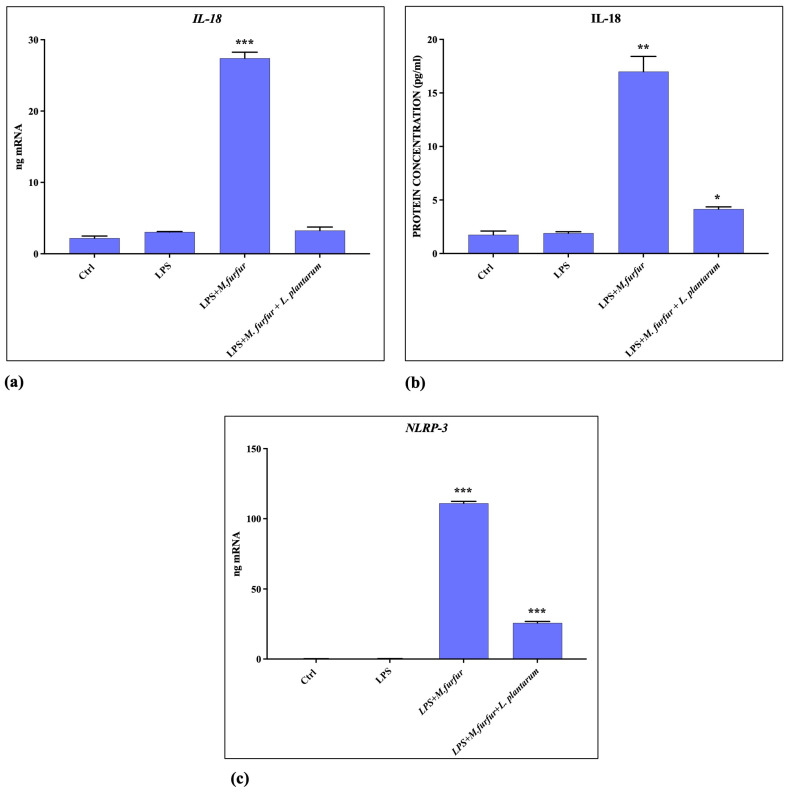
Inflammasome activation evaluation. mRNA (**a**,**c**) and protein (**b**) expression of IL-18 and NLRP-3 in HaCat cell pre-treated for 4 h with LPS of *S.* Typhimurium and infected with *M. furfur* alone or in the presence of *L. plantarum* after 24 and 48 h. The data are representative of three different experiments ± SD. Significant differences are indicated by * *p* < 0.05, ** *p* < 0.01, *** *p* < 0.001.

**Table 1 jof-09-01153-t001:** Primer’s sequences and amplification programs used in this study.

Gene	Primer’s Sequence	Conditions	Product (bp)
*AhR*	5′-ACCTACGCCAGTCGCAA-3′ 5′-CTGACGCTGAGCCTAAGAAC-3′	30″ at 95 °C, 30″ at 60 °C, 30″ at 72 °C for 40 cycles	200
*CyP1A1*	5′-TCCAGAGACAACAGGTAAAACA-3′ 5′-AGGAAGGGCAGAGGAATGTGAT-3′	15″ at 95 °C, 35″ at 60 °C, 35″ at 72 °C for 45 cycles	371
*Occludin*	5′-TCAGGGAATATCCACCTATCACTTCAG-3′ 5′-CATCAGCAGCAGCCATGTACTCTTCAC-3′	10″ at 95 °C, 45″ at 60 °C for 40 cycles	188
*Zonulin-1*	5′-AGGGGCAGTGGTGGTTTTCTGTTCTTTC-3′ 5′-GCAGAGGTCAAAGTTCAAGGCTCAAGAGG-3′	10″ at 95 °C, 45″ at 60 °C for 40 cycles	217
*Claudin-1*	5′-CTGGGAGGTGCCCTACTTTG-3′ 5′-ACACGTAGTCTTTCCCGCTG-3′	1″ at 95 °C, 30″ at 60 °C, 20″ at 72 °C for 40 cycles	128
*NrF2*	5′-ACCACCCACAACTTACTGCC-3′ 5′-GCCATAGGAGTATGGGGGAT-3′	5″ at 95 °C, 2″ at 60 °C, 5″ at 72 °C for 40 cycles	121
*MMP-2*	5′-TGACGGTAAGGACGGACTC-3′ 5′-TGGAAGCGGATTGGAAACT-3′	5″ at 94 °C, 7″ at 57 °C, 14″ at 72 °C for 40 cycles	342
*Caspase-1*	5′-GCCTGTTCCTGTGATGTGGAG-3′ 5′-TGCCCACAGACATTCATACAGTTTC-3′	15″ at 95 °C, 1′ at 60 °C for 40 cycles	165
*IL-1 β*	5′-GCATCCAGCTACGAATCTCC-3′ 5′-CCACATTCAGCACAGGACTC-3′	5″ at 95 °C, 9″ at 58 °C, 19″ at 72 °C for 40 cycles	708
*IL-18*	5′-GATTACTTTGGCAAGCTTGAA-3′ 5′-GCTTTCGTTTTGAACAGTGAA-3′	5″ at 95 °C, 6″ at 53 °C, 12″ at 72 °C for 40 cycles	470
*NLRP-3*	5′-GATCTTCGCTGCGATCAACA-3′ 5′-GGGATTCGAAACACGTGCATTA-3′	5″ at 95 °C, 34″ at 60 °C for 40 cycles	93
*β-ACTIN*	5′-CGTGGGCCGCCCTAGGCACCA-3′ 5′-TTGGCCTTAGGGTTCAGGGGGG-3′		243

## Data Availability

The authors confirm that the data supporting the findings of this study are available within the article.

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
