# Peer review of "Regulatory Ability of Lactiplantibacillus plantarum on Human Skin Health by Counteracting In Vitro Malassezia furfur Effects"

_jof, 2023, doi:10.3390/jof9121153_

Round 1

Reviewer 1 Report

Comments and Suggestions for Authors

as the attached file

Comments on the Quality of English Language

NO comments

Author Response

 Dear Reviewer,

 thank so much the referees for your valuable comments. As reported in the following, we answered all your questions and we've made all the suggested changes for re-submission of our paper.

Lactiplantibacillus plantarum is a widely distributed and versatile lactic acid bacterium. The authors shall describe the current understanding mechanism of probiotics on skin.

 Currently there are many works that focus on the importance of probiotics in different anatomical districts, and the indirect (through food intake) or direct effects on the skin. We have added in the text (lines 102-113) only some of the results reported in bibliography with the respective bibliographical references in the “References” section [5,6,27]

 All the figure legend shall be rewritten with clear and concise description. A brief title shall be described.

 Done.

Section 2.3 Phospholipase Assay L. plantarum and M. furfur were co-cultured in MRS broth at 30°C for 48 hours. Then, the mixture were plating on SDA agar containing b-endorphin for 10 days. Again, the yeast colonies were picked, transferred to egg-yolk agar for another 10 days. Although b-endorphin is known to stimulate phospholipase production by Malassezia, what is the purpose of plating the yeast on SDA agar containing b-endorphin? In Figure 1, it is hard to see the result of halo precipitation. Why not determine the phospholipase activity directly?

 The Egg-Yolk agar test is an assay used very frequently to detect the phospholipase activity of Malassezia spp., as widely reported in literature: Cafarchia et al., 2004, 2007, 2008, 2009; Machado et al., 2010; Vlachos et al., 2013; Honnavar et al., 2017; Charu Jain et al., 2017 (some of these references have been added into the manuscript). For this reason, we thought it was a valid test to repeat without having to resort to direct activity assays. The assay was repeated only in the experimental condition of co-culture of L.plantarum and Malassezia furfur so as to obtain an image with better resolution, replaced in Fig. 1 together with the previous one.

Figure 2: The HaCat cell were pretreated with L. plantarum for 2h. The description in figure legend is misleading. The similar descriptions were also found in Figure 4~8.

 We are sorry for the mistake. All inconsistencies have been corrected.

Line 204: In Table 1, table captions must be placed above the tables.

 Done.

Line 202: What is the real-time PCR condition for b-ACTIN?

 beta-actin is a housekeeping gene, and as such it is always expressed and amplified with any program. Therefore, there is no need to indicate the amplification conditions, as the program of the paired gene of interest is used for each run.

Line 437: S. Typhimurium vs. S. typhimurium

Typhimurium refers to serovar and is written in capital letters. It was our mistake to write it differently at the first mention, but we have corrected the name (line 153).

Line 543: zonulin e claudin vs. zonulin and claudin

Done.

Figure 7: Several mistakes were found in the figure legend. mRNA (a, c b) and protein (b c, d) expression. after 24 and 48h vs. After 4 h of infection.

 All mistakes were corrected.

Figure 8: Several mistakes were found in the figure legend. Where is the L. plantarum treated group? The figure of “a b c” were not cited. after 24 and 48h àAfter 4 h of infection

All mistakes were corrected.

Discussion: 1. The current discussion section appears to be insufficient in addressing the implications, limitations, and significance of the research findings. How about the effect of probiotics pretreatment of HaCat cell? How about co-treatment of L. plantarum and M. furfur?

 Obviously the limitation of our work is not having performed in vivo experiments, for this reason it seemed appropriate to underline it also in the title, which was modified by adding the word "in vitro". The use of pre-treatment in our experimental model was underlined in the discussion section (lines 585-599), while co-treatment was exclusively a requirement linked to the type of experiment, as reported in the cited reference 23, for inflammosome study, and in phospholipase activity assay.

 The authors shall discuss how the current findings align with or differ from the results of previous studies

Our results align perfectly with the current bibliography which demonstrates the ameliorative and anti-inflammatory effects of probiotics on the skin, the differences consist in the point of view in the research and obviously in the experimental model chosen. Our model is aimed at verifying a topical action of live probiotics and not of portions of them or culture supernatants in HaCat cells which are influenced by a saprophyte in some cases pathogenic such as Malassezia furfur (Discussion section lines 585-599)

Reviewer 2 Report

Comments and Suggestions for Authors

The introduction section, particularly the 1st paragraph, needs references, in order for the uninitiated to further seek information.

Too much is attributed to Lactiplantibacillus, and not necessarily supported by the literature. Anti-cancer properties for example is a rather frivolous statement, and its anti-colitis use has been the subject of judicial/ scientific debate, I believe? And do references 19-21, the ones used here, really mention this?

Also, the effect of general lactobacillus sp. on everything from diabetes to neuropsychiatric disorders, as stated in 99-102 lines, is again a frivolous statement, moreover since it is not supported from any literature references, you have not used any. 

In the discussion you state that M. furfur might be an excellent model for studying MPPRs etc. You give seven references for this, that are all yours. This denotes that you are studying this and believing this, you may be right but you have to state that no one else has done it, has experimented on this. 

Comments on the Quality of English Language

The manuscript needs some language polishing in order for the reader to follow it easily, the abstract too

Author Response

Dear Reviewer,

 thank so much the referees for your valuable comments. As reported in the following, we answered all your questions and we've made all the suggested changes for re-submission of our paper.

The introduction section, particularly the 1st paragraph, needs references, in order for the uninitiated to further seek information.

 We added some references to the paragraph as suggested [1-6].

 Too much is attributed to Lactiplantibacillus, and not necessarily supported by the literature. Anti-cancer properties for example is a rather frivolous statement, and its anti-colitis use has been the subject of judicial/ scientific debate, I believe? And do references 19-21, the ones used here, really mention this?

 There are many works on the beneficial effects of Lactobacilli, the limited bibliography reported is due to the fact that the focus of our work was not that, but only the interaction between Malassezia and L. plantarum on the skin. Welcoming your criticism we preferred to eliminate the description of the effects considered ” frivolous” from the text which are nevertheless reported in current bibliography of PubMed. For example, anti-cancer effects are referred to the action of probiotics on melanoma cells or on 293T cells, these effects are related to slowing down proliferation or blocking the cell cycle which the authors see as a positive factor in the problem of metastasis. (Park Jet al. Anti-Cancer Effects of Lactobacillus plantarum L-14 Cell-Free Extract on Human Malignant Melanoma A375 Cells. Molecules. 2020 Aug 26;25(17):3895; Liu J. et al. Exploring the Antioxidant Effects and Periodic Regulation of Cancer Cells by Polyphenols Produced by the Fermentation of Grape Skin by Lactobacillus plantarum KFY02. Biomolecules. 2019 Oct 6;9(10):575.)

 Also, the effect of general lactobacillus sp. on everything from diabetes to neuropsychiatric disorders, as stated in 99-102 lines, is again a frivolous statement, moreover since it is not supported from any literature references, you have not used any. 

Also in this case we welcoming your criticism and preferred to eliminate the description of the effects considered ” frivolous statement” because of it would not be useful for the purposes of our work to dwell on effects not directly related to our experimental model and to extend the bibliographic data already reported.

In the discussion you state that M. furfur might be an excellent model for studying MPPRs etc. You give seven references for this, that are all yours. This denotes that you are studying this and believing this, you may be right but you have to state that no one else has done it, has experimented on this. 

According as you underline we delete some of our reference and added some of other authors, but meanwhile the study of Malassezia spp ant its interaction with the host is one of our most important lines of research [36,37].

The manuscript needs some language polishing in order for the reader to follow it easily, the abstract too

The language editing has been performed by papertrue.com.

Round 2

Reviewer 1 Report

Comments and Suggestions for Authors

The paper can be accepted without any further changes.

Reviewer 2 Report

Comments and Suggestions for Authors

satisfied